# The Effects of Optical Aberrations to Illumination Beam Thickness in Two-Photon Excitation Microscopes

Nan Li [1,2], Fanglin Luo [1,2], Chengliang Yang [1,2], Zenghui Peng [1,2], Li Xuan [1,2], Qingpan Bu [3], Quanquan Mu [1,2,*] and Xingyun Zhang [1,2,*]

[1] State Key Laboratory of Applied Optics, Changchun Institute of Optics, Fine Mechanics and Physics, Chinese Academy of Sciences, Changchun 130033, China; linan@ciomp.ac.cn (N.L.); luofanglin20@mails.ucas.ac.cn (F.L.); ycldahai@ciomp.ac.cn (C.Y.); peng@ciomp.ac.cn (Z.P.); xuanli@ciomp.ac.cn (L.X.)

[2] Center of Materials Science and Optoelectronics Engineering, University of Chinese Academy of Sciences, Beijing 100049, China

[3] School of Life Sciences, Changchun Normal University, Changchun 130032, China; buqingpan@ccsfu.edu.cn

[*] Correspondence: muquanquan@ciomp.ac.cn (Q.M.); xyzhang@ciomp.ac.cn (X.Z.)

**Abstract:** When performing in vivo imaging of live samples, it is a big challenge to penetrate thick tissues while still maintaining high resolution and a large field of view because of the sample-induced aberrations. These requirements can be met by combining the benefits of two-photon excitation, beam modulation and adaptive optics in an illumination path. However, the relationship between aberrations and the performance of such a microscopy system has never been systematically and comprehensively assessed. Here, two-photon Gaussian and Bessel beams are modulated as illumination beams, and how aberrations affect the thickness of the illumination beams is evaluated. It is found that the thickness variation is highly related to the azimuthal order of Zernike modes. The thickness of the two-photon Gaussian beam is more sensitive to Zernike modes with lower azimuthal order, while the thickness of the two-photon Bessel beam is more sensitive to the higher-azimuthal-order Zernike modes. So, it is necessary to design a new strategy to correct aberrations according to the effects of different Zernike modes in order to maximize the correction capability of correctors and reduce the correction errors for those insensitive Zernike modes. These results may provide important guidance for the design and evaluation of adaptive optical systems in a two-photon excitation microscope.

**Keywords:** aberration analysis; beam modulation; two-photon excitation microscope; resolution

## 1. Introduction

The two-photon excitation microscope (TPM) has received much attention and has been rapidly developed since being proposed by W. Denk et al. in 1990 [1]. Compared with conventional confocal microscopy, TPM provides high spatial resolution imaging with lower amounts of scattering in deep regions of biological samples because of longer wavelength illumination and nonlinear excitation effects. However, the temporal resolution of TPM is limited, since a scanning system is indispensable in order to obtain a large field of view (FOV). In recent years, the Bessel beam (a kind of diffraction-free beam) has drawn much attention to be used as an illumination beam of TPM, because it can achieve high spatiotemporal resolution while maintaining large FOV. In 2013, F. O. Fahrbach et al. [2] developed a two-photon Bessel light-sheet microscope, and the light-sheet penetration depth was increased by a factor of 3–5 compared to linear excitation with a Gaussian beam. In 2020, Na Ji et al. [3] developed a Bessel two-photon laser scanning microscope for high-speed volumetric fluorescence imaging of neurovascular dynamics. The 22-times increase in the axial excitation range allowed them to use 2D scanning of the Bessel beam to image a volume, instead of the 3D scanning of the Gaussian beam.

Although using two-photon Bessel beam (TPBB) as illumination beam could provide many advantages, the performance of the microscopy system (especially resolution) is still dramatically affected by the sample-induced optical aberrations, as the illumination beam goes deep into the biological tissues [4]. Especially when imaging inside semi-transparent samples, the image quality is weakly affected by scattering up to a depth of several scattering lengths [5], so optical aberrations would be the main influence factor. Optical aberrations will lead to a spreading of the focusing spot inside the sample, both in axial and lateral directions, which will increase the thickness of TPBB and ultimately reduce the resolution of the microscope.

Adaptive optics (AO) is the most powerful technique to correct the aberrations of optical systems and has been widely used in Gauss beam illuminated two-photon microscopes [6–8] (TPGM) and light-sheet microscopes [9–11] for the past few years. However, there has been little research on integrating the AO elements in a two-photon Bessel microscope (TPBM) up to now. Because of the differences in principles [12], methods [13] and light path conjugation relationships [14] to generate the TPBB and two-photon Gaussian beam (TPGB), the sample-induced aberrations have totally different effects even if the illumination beam passes the same part of biological sample. Therefore, quantitative analysis of how aberrations affect the TPBB is rather important to design a reasonable AO system for TPBM.

In this paper, we analyzed the effects of optical aberrations to the thickness of TPGB and TPBB by simulating the practical optical system and imaging process. We found that the thickness variations of these two beams are both related to the azimuthal order of Zernike modes but have different effect models. TPGB is more sensitive to those Zernike modes with lower azimuthal order, while TPBB is more sensitive to the higher azimuthal-order Zernike modes. Moreover, it is necessary to design a new strategy to correct the aberrations according to the effects of different Zernike modes in order to maximize the correction capability of the corrector and reduce the correction errors for those insensitive Zernike modes. Finally, further reasonable advice on AO is provided.

## 2. Simulation Methods and Evaluation Parameters

### 2.1. Simulation Methods and Aberration Description

Generally, a conventional AO system employs a wavefront sensor to measure aberrations, an adaptive correction element to compensate aberrations and a control system that processes the sensor signals in order to drive the correction element [15]. Combining the microscope and AO system, the schematic diagram of the illumination path and direct wavefront detection path of AO-TPBM and AO-TPGM are shown in Figure 1, regardless of the scanning system. For simplicity, the correction element (generally spatial light modulator, SLM) is drawn as a transmission device rather than a reflective device in reality. In AO-TPGM, in most cases, the fs-NIR-laser beam propagates through a series of 4f lenses and then converges by the illumination objective (IO) to form a Gaussian focus in the sample. While in AO-TPBM, the illumination beam is modulated by the axicon phase pattern on SLM to form a Bessel beam firstly. Then, IO and other lenses constitute 4f systems together and deliver the modulated Bessel beam into the sample. The two-photon excitation fluorescent guide star (GS) generated at the sample is collected by IO and directed to a wavefront sensor (WFS). Therefore, these two kinds of systems are very different in the conjugation relationship among SLM, WFS, IO and the sample. In AO-TPGM, SLM and WFS are both conjugated with the rear pupil of IO. In AO-TPBM, WFS is conjugated with the rear focal plane of IO, while SLM is conjugated with the sample.

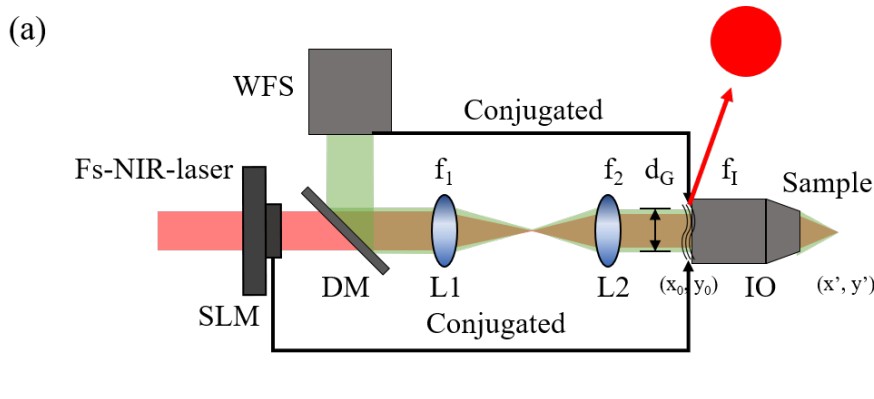

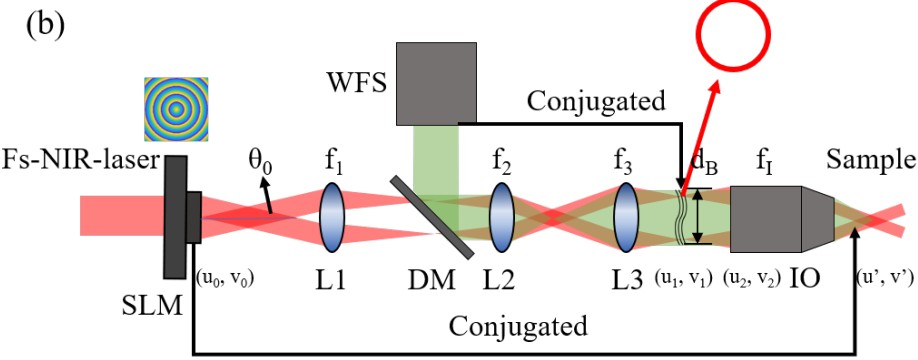

**Figure 1.** The schematic diagram of the illumination path and direct wavefront detection path of (**a**) AO-TPGM and (**b**) AO-TPBM without scanning system. The intensity profile of the illumination beam at WFS-conjugated plane is given.

During AO correction, SLM is controlled with the wavefront measured by WFS. For the sake of convenience, generally the wavefront aberration is decomposed into Zernike modes. In our simulation systems, the same method is used to simulate the aberrations caused by samples. The aberrations conjugated with WFS can be expressed as [16]:

$$W(x,y) = \sum_j a_j Z_j(x,y) \tag{1}$$

where $a_j$ is the coefficient that describes the amplitude of Zernike mode $Z_j(r, \theta)$. With $j = (n(n+2)+m)/2$, the Zernike modes are defined as:

$$Z_n^m(r,\theta) = \begin{cases} m < 0 : \sqrt{2(n+1)} R_n^{-m}(r) \sin(-m\theta) \\ m = 0 : \qquad \sqrt{n+1} R_n^0(r) \\ m > 0 : \sqrt{2(n+1)} R_n^m(r) \cos(m\theta) \end{cases} \tag{2}$$

$$R_n^m(r) = \sum_{s=0}^{(n-m)/2} \frac{(-1)^s (n-s)!}{s!((n+m)/2 - s)!((n-m)/2 - s)!} r^{n-2s} \tag{3}$$

The indices n and m represent the radial and azimuthal order, respectively. They are restricted to the conditions that $n \geq |m|$ and $n - |m|$ is even. The relationship between polar and rectangular coordinates is as follows:

$$x = r\cos\theta, \ y = r\sin\theta, \ \text{and} \ 0 \leq r = \sqrt{x^2 + y^2} \leq 1 \tag{4}$$

For TPGB, the illumination beam at the pupil of IO can be defined:

$$E_{G0}(x_0, y_0) = A_{G0}\exp(-\frac{x_0^2 + y_0^2}{w_{G0}^2}) \tag{5}$$

where $w_{G0}$ is the waist radius of Gaussian beam and $A_{G0}$ is the amplitude. Thus, the amplitude distribution of the aberrated beam behind IO becomes:

$$E_{GI}(x_0, y_0) = E_{G0}(x_0, y_0)\exp[-i2\pi W(x_0, y_0)]t_I(x_0, y_0) \tag{6}$$

$t_I$ is the transmittance function of IO (with focal length $f_I$):

$$t_I(x_0, y_0) = \exp[-i\frac{k}{2f_I}(x_0^2 + y_0^2)] \tag{7}$$

Based on the law of Fresnel diffraction, the Gauss beam behind IO at distance z is:

$$\begin{aligned} E_G(x', y', z) &= \frac{\exp(ikz)}{i\lambda z}\iint E_{GI}(x_0, y_0)\exp\{i\frac{k}{2z}[(x' - x_0)^2 + (y' - y_0)^2]\}dx_0dy_0 \\ &= Z\{E_{GI}(x_0, y_0), z\} \end{aligned} \tag{8}$$

Finally, the two-photon excited fluorescence intensity is defined as the fourth power of amplitude:

$$I_G(x', y', z) = |E_G(x', y', z)|^4 \tag{9}$$

For TPBB, in order to modulate the Bessel beam, an axicon phase pattern is generated by SLM:

$$\Phi_B(u_0, v_0) = \frac{2\pi\sqrt{u_0^2 + v_0^2}}{r_0} \tag{10}$$

Based on Vasara A's research [13], the modulation parameter $r_0 = 2\pi/\alpha = 2\pi/k\theta_0$ is the width of the concentric ring in this pattern, where $k = 2\pi/\lambda$ and $\lambda$ is the wavelength of the illumination laser. $\theta_0$ is the oblique angle of equiphase surface shown in Figure 1b, and $\alpha$ represents the propagation vector of the plane wave components of the angular spectrum to the transverse plane. If SLM is illuminated uniformly with Gaussian beam, this pattern produces a complex amplitude:

$$E_{B0}(u_0, v_0) = A_{B0}\exp(-\frac{u_0^2 + v_0^2}{w_{B0}^2})\exp[-i\Phi_B(u_0, v_0)] \tag{11}$$

where $w_{B0}$ is the waist radius of input beam and $A_{B0}$ is the amplitude. Since SLM is conjugated to the sample plane, the complex amplitude at the rear focal plane of IO can be calculated by its 2D Fourier transform [17]:

$$\begin{aligned} E_{B1}(u_1, v_1) &= F\{E_{B0}(u_0, v_0)\} \\ &= \iint E_{B0}(u_0, v_0)\exp[-i2\pi(u_1u_0 + v_1v_0)]du_0dv_0 \end{aligned} \tag{12}$$

Thus, the complex amplitude of the illumination beam through the aberrations can be expressed as

$$E_{B2}(u_1, v_1) = E_{B1}(u_1, v_1)\exp[-i2\pi W(u_1, v_1)] \tag{13}$$

Because $E_{B2}(u_1, v_1)$ is located at a distance $f_I$ in front of IO, the complex amplitude immediately behind IO can be written as:

$$E_{BI}(u_2, v_2) = Z\{E_{B2}(u_1, v_1), f_I\}t(u_2, v_2) \tag{14}$$

In the same way, to find the distribution behind IO at distance z, Equation (8) is applied:

$$E_B(u', v', z) = Z\{E_{BI}(u_2, v_2), z\} \tag{15}$$

Finally, the two-photon excited fluorescence intensity is defined as the fourth power of amplitude:

$$I_B(u', v', z) = \left| E_B(u', v', z) \right|^4 \tag{16}$$

It is worth noting that the simulated aberration should be set at the plane, which is conjugated to WFS, and the diameter of the aberration should be equal to the diameter of the illumination beam at this plane. For TPGB, the diameter of the aberration (d) is:

$$d_G = 2w_{G0} \tag{17}$$

For TPBB, according to the geometric relation of the optical path, the diameter of aberration is equal to the diameter of the annular intensity distribution at the rear pupil of IO:

$$d_B = 2\theta_0 f_1 \frac{f_3}{f2} = \frac{2\lambda f_1 f_3}{r_0 f_2} \tag{18}$$

Based on Fresnel diffraction and Fourier Optics, we have derived the calculation process for TPGB and TPBB under the influence of aberration. With it, theoretical analysis of the effects of single Zernike mode and random aberrations can be carried out.

### 2.2. Evaluation Factors of Thickness

In order to describe the effect of the aberrations, we define a quantitative evaluation factor as mentioned in our previous article [18,19]. A relative factor TR is used to describe the thickness:

$$TR = \frac{T_A}{T_0} \tag{19}$$

$T_0$ and $T_A$ represent the thickness of aberration-free and aberrated beams, respectively, as shown in Figure 2a,b. Because of the distorted intensity distribution of the aberrated illumination beam, the full width at half maximum (FWHM) could not accurately describe the beam thickness. Thus, we define the thickness as minimum width that contains 63% (i.e., $1 - 1/e$) of the total beam's power [20].

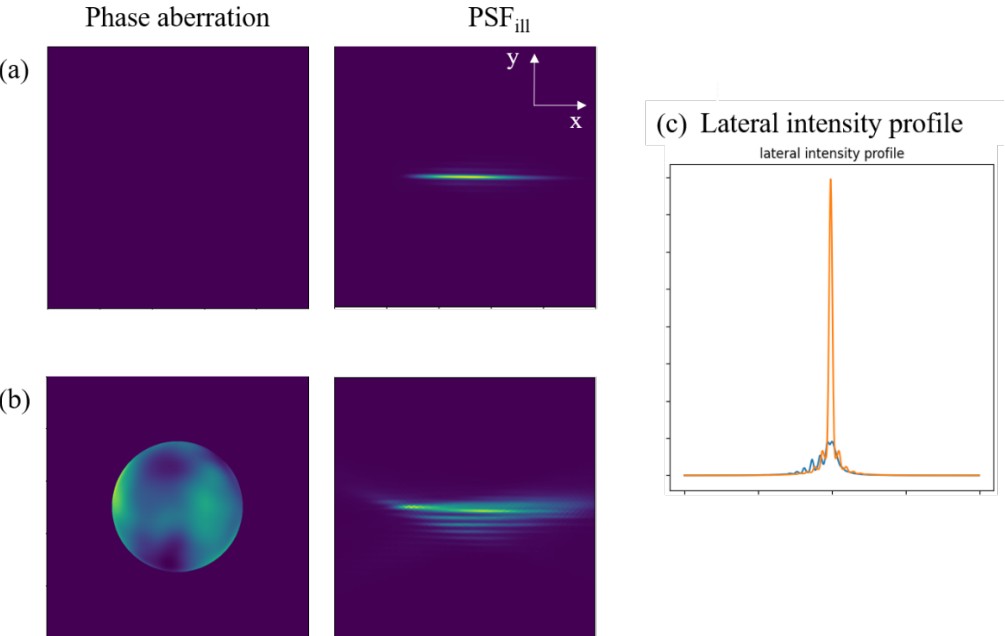

**Figure 2.** The TPBB simulation: (**a**) aberration-free phase and PSF$_{ill}$; (**b**) aberrated phase and PSF$_{ill}$; (**c**) lateral intensity distribution of the aberration-free and aberrated TPBB.

## 3. Results and Discussions

### 3.1. Effects of Single Zernike Mode Aberration

The first 104 Zernike modes and their distortion effects along the direction of illumination are simulated. For each mode, a series of amplitudes from $0.1\lambda$ to $2\lambda$ (PV) are simulated. For simplicity, only the first 14 Zernike modes with $1.0\lambda$ (PV) and the distorted PSF in XY plane are shown in Figure 3. Figure 4 gives the effect on TR of different Zernike modes with four sets of PVs ($0.3\lambda$, $0.8\lambda$, $1.3\lambda$ and $1.8\lambda$). According to these results, we can summarize the general rules of the aberrations' effect on TPGB and TPBB.

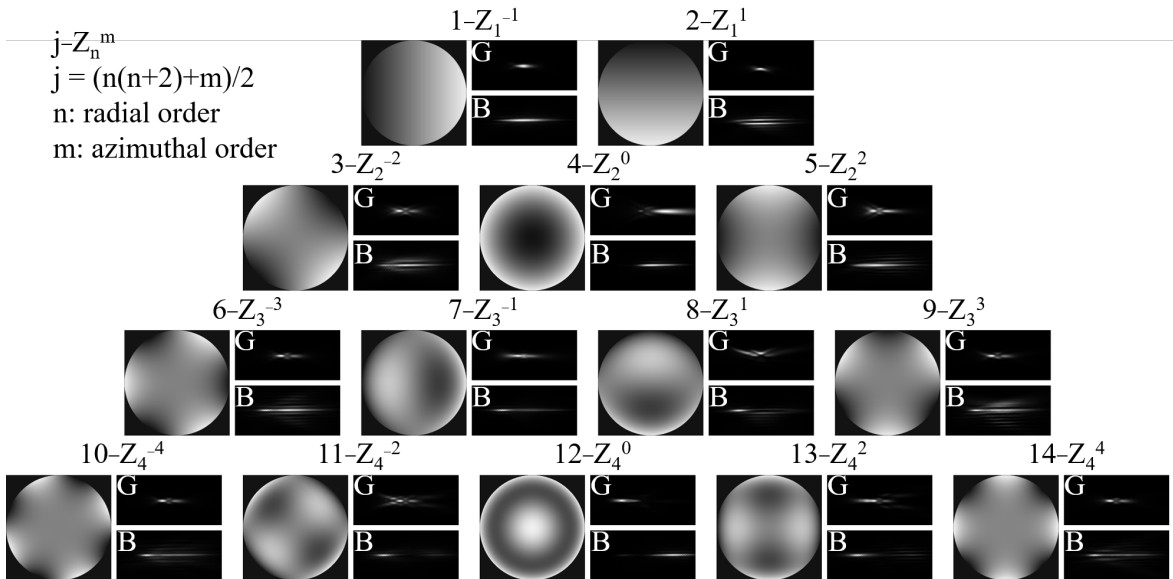

**Figure 3.** The first 14 Zernike modes and the distorted PSF along the direction of illumination of TPGB (G) and TPBB (B). All results are calculated for amplitude of $1.0\lambda$ (PV).

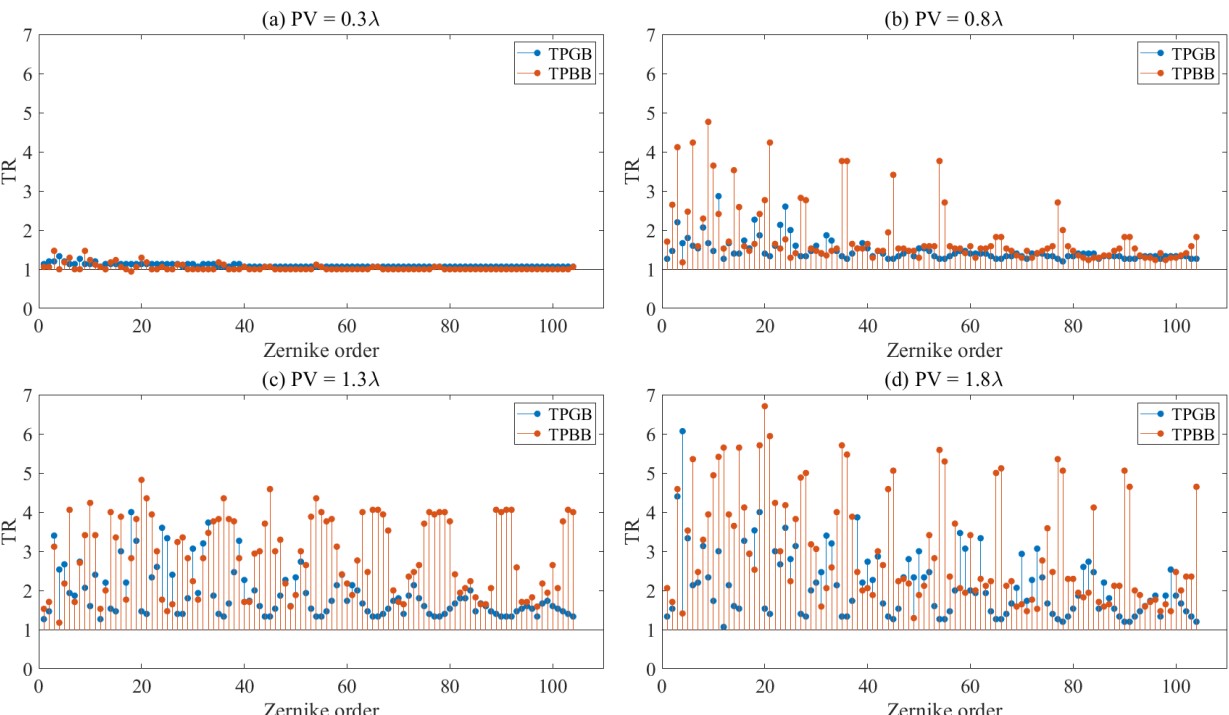

**Figure 4.** Relationship between TR and each Zernike mode for TPGB and TPBB. Amplitude of each mode is from $0.3\lambda$ to $1.8\lambda$.

For both TPGB and TPBB, as the amplitude of aberration increases, the TR becomes larger, i.e., the effect becomes more serious. The effects on TR of lower-order modes are greater than that of higher-order modes under relatively small amplitude (TPGB: PV < 1.3λ; TPBB: PV < 0.8λ). For the higher-order Zernike modes, the TPBB's TR would reach to a rather high level under smaller amplitude compared to TPGB.

What is more interesting is that the Zernike modes with the same azimuthal order have different effects on the thickness of TPGB and TPBB, as shown in Figure 5. Generally, the Zernike modes with lower azimuthal orders (m ≤ 4) have relatively stronger effects on the thickness of TPGB, while they have smaller effects on thickness of TPBB. In contrast, the Zernike modes with higher azimuthal orders (n − |m| ≤ 4) have relatively stronger effects on the thickness of TPBB, while they have smaller effects on the thickness of TPGB.

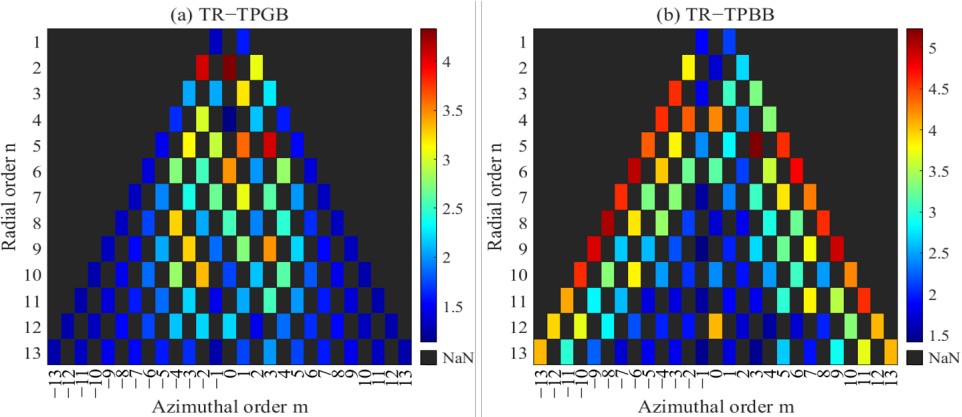

**Figure 5.** (**a**) TPGB and (**b**) TPBB are sensitive to the Zernike modes with different azimuthal orders. Amplitude (PV) of each mode is 1.6λ.

In order to explain this difference, we go back to the definition of Zernike modes. According to Equation (3), there are fewer lower-order terms of r in $R_n^m(r)$ as the increasing of azimuthal order m. Figure 6 shows how a radial circle polynomial $R_n^m(r)$ varies with r and m. When the radial order is odd, the fluctuation of the aberrated wavefront gradually shifts from the center to the edge as the azimuthal order increases. The same thing happens when radial order is even; however, the central fluctuation is most dramatic when the azimuthal order is 2 rather than 0. The intensity profile of TPGB and TPBB at the plane conjugated to the WFS are solid circle and hollow ring, respectively. Therefore, TPGB is more sensitive to the central fluctuation of the wavefront, and TPBB is more sensitive to the fluctuation in the periphery of the wavefront. Thus, TPGB is more sensitive to the Zernike modes whose azimuthal orders are relatively lower, especially less than 4. In contrast, those Zernike modes whose difference between radial order and azimuthal order is less than 4 have greater influence on the thickness of TPBB.

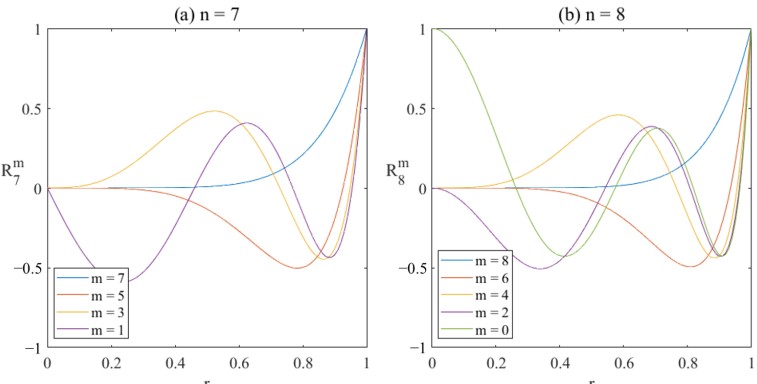

**Figure 6.** Variation of a radial circle polynomial $R_n^m(r)$ with r. (**a**) n = 7; (**b**) n = 8.

Another point worth noting about the Bessel beam is its 'self-reconstruction' property [20]. As shown in Figure 7, TPBB will gradually restore to its original shape as the aberration amplitude increases. Generally, the 'self-reconstruction' process takes place when the amplitude changes from 0.8λ to 1.5λ. For the Zernike modes that TPBB is sensitive to, at first TR is stabilized at about 4, and then it continues to increase after 1.6λ. For other Zernike modes, TR increases first and then declines as TPBB gradually reverts to its original shape. That is to say, the growth curve of TR is not monotonic but fluctuant.

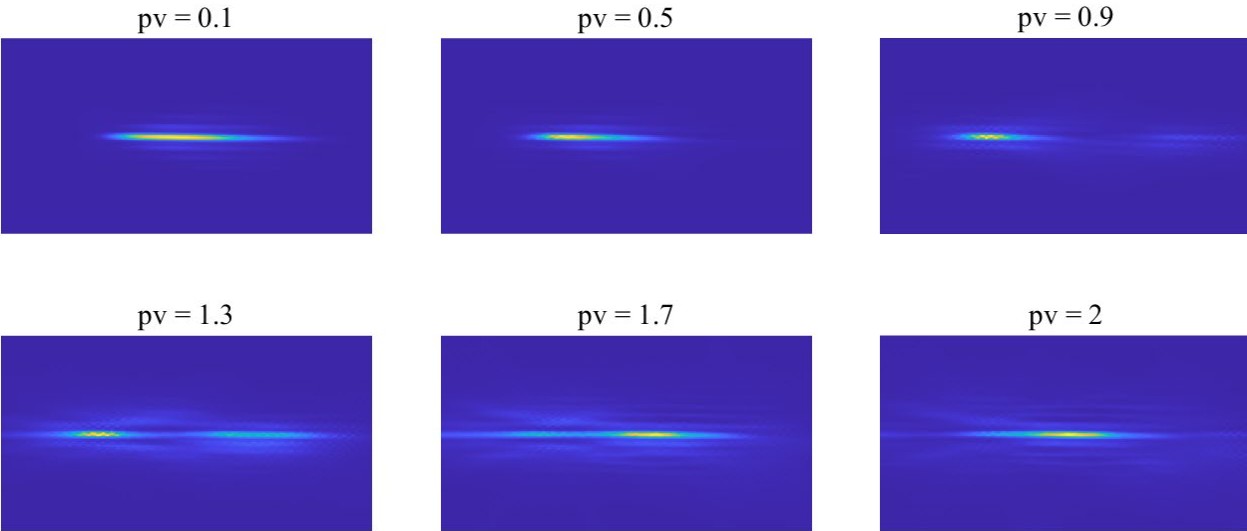

**Figure 7.** The illustration of the 'self-reconstruction' property of TPBB.

According to the above results, researchers in the field of AO should pay more attention to the different affection models of Zernike aberration to TPGB and TPBB. Considering all Zernike modes with the same weights would be a waste of the correction capability of the correction devices. For a Zernike mode which has less effect on thickness even with large amplitude, the corrector may consume a large amount of modulation to get a poor correction performance, which may even lead to a large correction error. Hence, it would be more efficient to develop a suitable strategy to filter out the Zernike modes to be corrected for different illumination beams. For TPBB, Zernike modes with higher azimuthal order should be considered with higher weights.

### 3.2. Effects of Random Biological Aberrations

In this section, we mainly focus on the effects of biological aberrations and the AO correction strategy to restore the quality of TPGB and TPBB. At first, it is necessary to understand the characteristics of biological aberrations and then simulate a realistic biological aberrations model. According to M. Schwertner and M. J. Booth et al. [21,22], aberration caused by the variation of biological specimen refractive index can be considered random. Moreover, the amplitudes of higher-order Zernike modes are usually smaller than those of the lower-order ones. Furthermore, according to the experimental results of Kai Wang et al. [6,7] on zebrafish embryo and mouse cortex, the first 65 Zernike modes are sufficient to describe biological aberrations. For superficial specimen (<0.1 mm), the aberration of biological sample is small, and RMS is approximately 0.1λ. As the illumination position goes deeper, the aberration would increase gradually to even larger than 0.5λ (RMS).

Thus, we use RMS to indicate the intensity of random aberrations caused by sample at different depths, and the variation ranges from 0.1λ to 1λ. Based on the aberration models mentioned above, each aberration intensity randomly generates 100 wavefronts, and the coefficients' distribution of each Zernike mode is shown in Figure 8. The coefficients of tip, tilt and defocus modes are set to 0, since those modes only affect the position of the illumination beam and have no effects on the thickness.

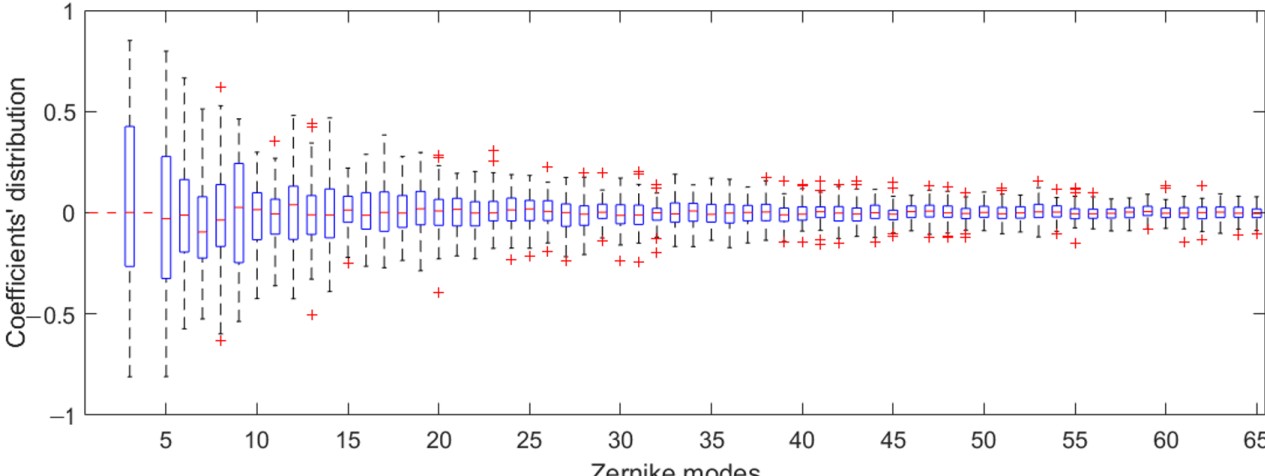

**Figure 8.** The coefficients' distribution of each Zernike mode in the model of sample-induced aberrations, where the coefficients of tip, tilt and defocus modes are set to 0. On each box, the central mark indicates the median, and the bottom and top edges of the box indicate the 25th and 75th percentiles, respectively. The whiskers extend to the most extreme data points not considered outliers, and the outliers are plotted individually using the '+' marker symbol.

The average TR of 100 groups of wavefronts is used to obtain the statistical results, as shown in Figure 9. For TPGB, the relationship between TR and RMS of biological aberration is nearly linear. For TPBB, the relationship is totally nonlinear. TR increases dramatically at first and then remains almost stable due to the 'self-reconstruction' property. When RMS is less than 0.4λ, the effect on the thickness of TPBB is stronger than that of TPGB, while the effect is far more serious for TPGB, as RMS changes from 0.5λ to 1.0λ. This interesting result shows that the effect of random aberrations to TPBB is restrained by the 'self-reconstruction' property. So, it would be better to use the fluorescent guide star (GS) excited by TPBB in the deeper specimen because the cross-section size of the GS is relatively smaller than that excited by TPGB, and the WFS can measure the wavefront more exactly. Of course, it should be a comprehensive consideration to choose which one as GS because the length of GS is another important factor.

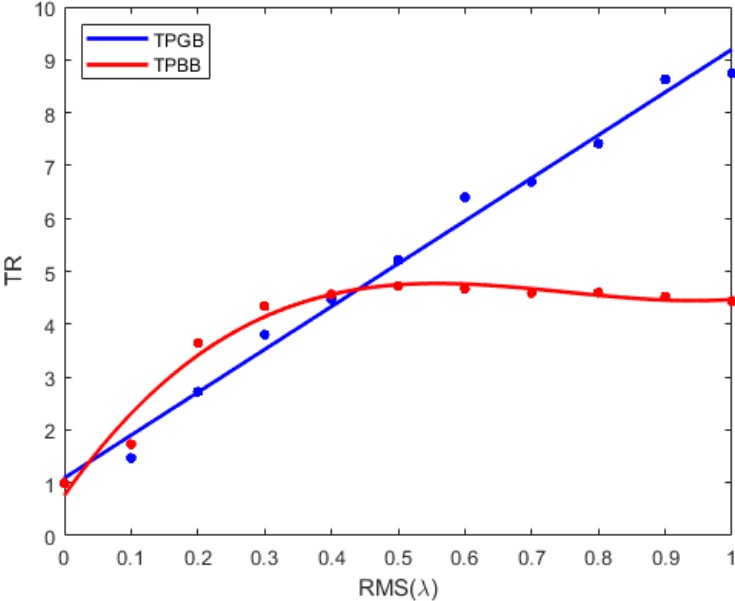

**Figure 9.** Relationship between TR and the random aberrations for TPBB and TPGB. Amplitude (RMS) is from 0.1λ to 1λ.

In order to determine how many Zernike modes are needed to correct to eliminate the distortion caused by biological aberrations, we set the coefficient of each Zernike mode to zero in turn to simulate the conventional AO correction process. The closer to 1 the TR values are, the better the correction performance is. The relationship between TR and the number of corrected Zernike modes under random aberration with variable RMS is shown in Figure 10. If we assume that the correction of the AO system is ideal when TR is smaller than 1.1, we find very similar results for TPBB and TPGB. For small aberrations (RMS $\leq 0.1\lambda$), the first 14 modes should be corrected to achieve ideal correction. For moderate aberrations (RMS $\leq 0.3\lambda$), they both need to correct the first 35 modes. For large aberrations (RMS $\geq 0.3\lambda$), more than 50 modes should be corrected. However, the decreasing trend of TPGB's TR is smoother compared with that of TPBB with the increase in corrected Zernike modes. This is because the Zernike modes with higher azimuthal orders have a stronger effect on the thickness of TPBB. Therefore, there are some steps in the decreasing trend curves. Once again, this step-changing trend proves that it is necessary to use a kind of filter function to correct different Zernike modes with different weights for TPBB in order to maximize the correction capability of the corrector and reduce the correction errors for those insensitive Zernike modes.

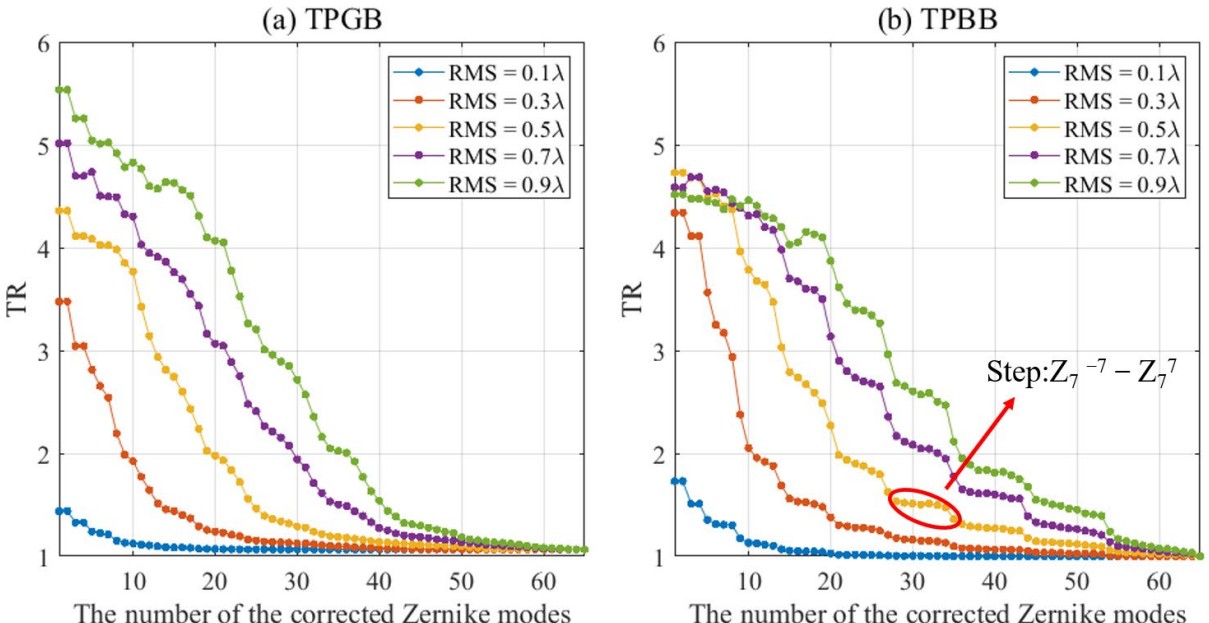

**Figure 10.** TR of different RMS of the initial random aberration varying with the number of corrected Zernike modes for (**a**) TPGB and (**b**) TPBB.

## 4. Conclusions

In this paper, two-photon excitation microscopy systems combined with AO using TPGB and TPBB as illumination beams are simulated. The effects of each single Zernike mode and random biological aberration to TPGB and TPBB are investigated and compared.

Firstly, aberrations induced by samples will make the illumination beam thicker than the ideal situation, and lower-order Zernike modes generally have greater effects on the thickness than higher-order modes. TPGB is more sensitive to the Zernike modes with a lower azimuthal order ($|m| \leq 4$), while TPBB is more sensitive to the Zernike modes with a higher azimuthal order ($n - |m| \leq 4$). Secondly, because of the 'self-reconstruction' property, the relationship between thickness of TPBB and the amplitude of wavefront aberration is not monotonically or linearly increasing like TPGB. When the aberration is larger than a certain value, the thickness may not increase anymore, or it may even decrease. Finally, the number of Zernike modes that need to be corrected under different aberration situations is given. For small aberrations (RMS $\leq 0.1\lambda$), the first 14 modes should be corrected; for moderate aberrations (RMS $\leq 0.3\lambda$), the first 35 modes should be

corrected; and for large aberrations (RMS $\geq 0.3\lambda$), more than 50 modes should be corrected. These results will contribute to the design of AO systems and the performance evaluation of microscopy systems. On the basis of the above conclusions, it is necessary to further explore a kind of filter function to correct different Zernike modes with different weights for different illumination beams in order to maximize the correction capability of the corrector and reduce the correction errors for those insensitive Zernike modes. We will study this in the future.

One of the main purposes of this article is to present a simulation method for two-photon excitation microscopes. With this method, we could simulate the practical optical system and calculate the intensity and phase at planes conjugated to WFS, SLM and other vital devices using Fourier angular spectrum diffraction theorem and Fresnel diffraction. Then, we could analyze the changes in system performance caused by various factors, such as aberrations. This method is not restricted to analyzing the thickness of TPGB and TPBB. It is a universal method to simulate and analyze the effects of aberrations to one- or multi-photon Gaussian beams, Bessel beams, Airy beams and so on.

**Author Contributions:** Conceptualization, N.L.; Data Curation, N.L. and X.Z.; Formal Analysis, N.L.; Funding Acquisition, Q.B. and X.Z.; Investigation, N.L. and F.L.; Methodology, N.L.; Project Administration, Q.M.; Resources, N.L. and X.Z.; Software, N.L. and X.Z.; Supervision, C.Y., Z.P., L.X. and Q.M.; Validation, F.L. and Q.M.; Visualization, N.L.; Writing—Original Draft, N.L.; Writing—Review and Editing, N.L., Q.M. and X.Z. All authors have read and agreed to the published version of the manuscript.

**Funding:** The research was funded by the Scientific Instrument Developing Project of the Chinese Academy of Sciences, grant number YJKYYQ20210037, the Science and Technology Development Program of Changchun, and the Natural Science Foundation of the Jilin Province Department of Education, grant number JJKH20220833KJ.

**Institutional Review Board Statement:** Not applicable.

**Informed Consent Statement:** Not applicable.

**Data Availability Statement:** The data that support the findings of this study are available from the corresponding author upon reasonable request.

**Acknowledgments:** Nan Li thanks the CAS Interdisciplinary Innovation Team for support for this work.

**Conflicts of Interest:** The authors declare no conflict of interest.

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
