# Peer review of "The Effects of Optical Aberrations to Illumination Beam Thickness in Two-Photon Excitation Microscopes"

_applsci, doi:10.3390/app12147156_

Round 1
Reviewer 1 Report
Two-photon excitation microscope (TPM) has recently raised great attention since it can observe deep tissues. Analyzing the effect of optical aberrations induced by sample on microscope performance is essential to improve TPM. This manuscript systematically analyzes the influence of different optical aberrations on the thickness of two-photon Bessel beam (TPBB) and two-photon Gaussian beam (TPGB). The content is well organized, and the simulation result is well presented. This manuscript deserves publication. The only required revision is to ensure the font size consistent in the plot and text.
Reviewer 2 Report
This is a rather technical, yet important paper presenting an original research devoted to the improvement of the quality of 2-photon imaging. I recommend acceptance of this paper, which suits well for Applied Sciences.
Some minor points:
1) Please check the numbering of the equations within the text
2) "tip, tilt and defocus" - add comma: "tip, tilt, and defocus"
3) add spaces after (a), (b), and (c) (i.e. Figure 2)
Reviewer 3 Report
This theoretical contribution presents a strategy to correct optical aberrations in two-photon excitation microscopy in live tissues (in vivo) using adaptive optics. The corrections are done to illumination beam thickness. Two types of beams are considered: two-photon Gaussian and Bessel beams. The authors found that the thickness variation is related to the azimuthal order of Zernike modes. Some conclusions are drawn and practical advises are given.
Science is sound and correct throughout the manuscript, the content is novel, while language, style and text organization are excellent. References are well chosen. In spite of that, there are errors in the submission that must be dealt with. I believe the article will be publishable after these errors are removed. The main error is the first from the list below and failing to remove it would be eliminatory and would lead to rejection.
I hope that the authors will find the suggested correction both helpful and easy to implement.
Below are my point-by-point suggestions for the manuscript corrections.
1. The main problem is that some sentences are copied verbatim from other scientists’ works published in other journals. Correcting these is MANDATORY! Please rephrase/rewrite ALL sentences of which you are not the authors, since what you did can be (quite needlessly) regarded as plagiarism and disqualify the whole work. The copied sentences I found include “Due to the nonlinear excitation with femtosecond near-infrared laser pulses, fluorescence is exclusively generated at the laser focus,” copied from Grewe et al, Biomed Opt Express, doi: 10.1364/BOE.2.002035, and “Thus, TPM provides high spatial-resolution imaging in deep region of biological samples, owing to lower amounts of scattering and out-of-focus fluorescence,” copied from Matsumoto et al, Sci. Rep. doi: 10.1038/s41598-018-27693-7. There was no need to use other peoples’ texts, as the article is quite well written anyway. I could not have searched manually for all possibly copied sentences throughout the manuscript so I cannot know if there are more such problematic places. If you did use some other sentences by other authors, please do correct them all and do not let that spoil the otherwise excellent text.
2. Main findings are not included in Abstract. Please add one sentence (in the line 20) on the influence of low- versus high-order Zernike modes to Gaussian and Bessel beams, since it is mandatory to include main results in Abstract.
3. Correct the first sentence of Abstract. It uses Latin expression “in vivo” as if it were English. Replace “In vivo imaging of live samples, it is a big challenge” with proper “When performing in vivo imaging of live samples, it is a big challenge”
4. Also in Abstract, delete the word “respectively” since it is nonsensical in the context it was used.
5. English and style are excellent throughout and only some minor errors were made. For instance, do not start a sentence (or a paragraph!) with “Where” or “While,” since these represent a continuation of the previous sentence/statement. There are several places where this error occurs.
6. In line 60, replace the sentence “fs-NIR-laser is propagated through a series of 4f lenses” by “fs-NIR-laser beam propagates through a series of 4f lenses”
7. Could you rephrase the expression “rare pupil”?
8. In line 116, replace “gauss” with “Gauss”
9. In line 222, replace “is stabled” by “is stabilized”
10. Summarize the part around “We will further study on it in the future” and introduce it to Conclusion (which is a proper place to speak about future work).
11. From “considered to be” please remove “to be” since it is superfluous.
12. Instead of writing “um” write “mm”, i.e. use the notation for micrometer correctly, with the Greek letter micro (m).
13. In line 266, what does the word “affection” mean? Please reconsider and rewrite.
14. Replace “how many Zernike modes is needed” by “how many Zernike modes are needed”
15. In Conclusion, replace “In this paper, microscopy systems” by “In this paper, two-photon excitation microscopy systems”
16. In Line 307, replace “the thickness may not increase anymore or even decrease” by “the thickness may not increase anymore or it may even decrease”
